behaviour, ecology

species redistributions, tropical vagrant fish, ocean warming, boldness, behavioural trade-offs

**Author for correspondence:**
Ivan Nagelkerken
e-mail: ivan.nagelkerken@adelaide.edu.au

# Coral-reef fishes can become more risk-averse at their poleward range limits

Ericka O. C. Coni[1], David J. Booth[2] and Ivan Nagelkerken[1]

[1]Southern Seas Ecology Laboratories, School of Biological Sciences, and The Environment Institute,
The University of Adelaide, Adelaide, SA 5005, Australia
[2]Fish Ecology Lab, School of Life Sciences, University of Technology Sydney, Ultimo, NSW 2007, Australia

DJB, 0000-0002-8256-1412; IN, 0000-0003-4499-3940

As climate warms, tropical species are expanding their distribution to temperate ecosystems where they are confronted with novel predators and habitats. Predation strongly regulates ecological communities, and range-extending species that adopt an effective antipredator strategy have a higher likelihood to persist in non-native environments. Here, we test this hypothesis by comparing various proxies of antipredator and other fitness-related behaviours between range-extending tropical fishes and native-temperate fishes at multiple sites across a 730 km latitudinal range. Although some behavioural proxies of risk aversion remained unaltered for individual tropical fish species, in general they became more risk-averse (increased sheltering and/or flight initiation distance), and their activity level decreased poleward. Nevertheless, they did not experience a decline in body condition or feeding rate in their temperate ranges. Temperate fishes did not show a consistently altered pattern in their behaviours across range locations, even though one species increased its flight initiation distance at the warm-temperate location and another one had lowest activity levels at the coldest range location. The maintenance of feeding and bite rate combined with a decreased activity level and increased sheltering may be behavioural strategies adopted by range-extending tropical fishes, to preserve energy and maintain fitness in their novel temperate ecosystems.

## 1. Introduction

A central tenet of biological invasions is that invaders create novel species interactions in recipient communities [1,2]. Climate change has intensified this phenomenon by facilitating species dispersion to regions where they did not occur historically and therefore these species need to adapt to survive under these new local conditions [3,4]. Marine animals exhibit faster range extensions than terrestrial organisms due to characteristics such as high propagule production and distant dispersal by ocean currents [5,6]. For example, tropical fishes are among the fastest organisms to shift their distribution to higher latitudes, facilitated by increased ocean temperatures and strength of major ocean currents such as Australia's East Australian Current [7,8]. The recruitment of tropical fish species has progressively increased in temperate ecosystems around the world, for example in southeast Australia, which is a hotspot of ocean warming and tropicalization. Yet, we do not fully understand potential mechanisms that might limit or facilitate these species to succeed in these new environments. Survivorship of tropical species is still low at higher latitudes due to detrimental minimum winter temperatures, but as the climate continues to warm permanent establishment will become more likely for some species [9,10]. Biological factors such as novel habitats, prey, predators and species interactions are also important [11,12] but remain largely unstudied.

Species that show adaptability in their antipredator strategies have a higher chance of survival in changing ecological environments [13]. Prey species that expand their range to novel environments under climate change are relieved

from historically important predators, but at the cost of gaining novel ones [2]. For example, tropical fish species might be particularly vulnerable to a native-temperate predator not only because of the lack of eco-evolutionary predator–prey experiences [14,15] but also because younger fishes are at a greater relative risk of predation [16,17]. Such lack of previous experiences and the unknown surrounding environment at temperate locations may induce neophobic predator avoidance in recruits and juvenile tropical fishes [18]. Although neophobia can constrain their establishment by reducing foraging opportunities, it would also reduce the potential detrimental costs of the 'initial encounter' with a novel predator in a novel environment [19]. Therefore, learning to assess the risk conditions of unfamiliar environments [20] and respond appropriately to novel temperate predators will allow tropical range-extending species to exhibit more efficient predator escape performance behaviour and facilitate their establishment in altered or suboptimal abiotic conditions [21].

Temperature is a major environmental determinant of life-history processes and governs basic physiological functions and behavioural traits, including predator evasion [22,23]. Temperature could affect antipredator responses through changes in swimming performance [10,24], muscle development [25], contractile properties of the swimming muscles [26] and neural control [22]. In addition to temperature effects, local habitat structure and composition can influence antipredator defences, with species that are able to adapt to novel surroundings and their associated threats having an increased advantage in persisting in novel habitats [27,28]. Fishes rely strongly on olfactory cues to perceive the presence and intensity of predation threat, but unfamiliarity with olfactory cues of novel environment may make predation risk hard to assess [29,30]. Thus, unfamiliarity with novel temperate habitats and predators combined with the physiological effects of low temperatures can significantly alter antipredator performance of tropical fishes in their novel temperate ranges.

Effective antipredator behaviours rely on many factors related to perceived risk and costs or benefits of escaping, such as predator size relative to prey, social interaction and proximity to refuge [31,32]. Escape responses often depend on the behavioural–environmental context [31]. Despite some differences among studies and species-specific responses [33,34], social species often have some advantage in their antipredator behaviour compared to solitary species, either because they feel safer through their shoaling behaviour and thus tolerate a closer predator approach, or initiate an escape response at greater distances because they are better at detecting approaching threats as a group [35–37]. Hence, range-extending species that shoal are more likely to persist during the initial stages of range extensions than solitary species [38], as they can learn to recognize unfamiliar predators from more experienced co-shoaling conspecific or native species [39,40]. Habitat context, such as refuge proximity, is an additional driver of antipredator behaviour. With decreasing distance to refuge, individuals become more confident and allow closer predator approaches [41,42]. While changes in temperature [43] and habitat context [44] might make early stages of coral reef fishes more vulnerable to predation in temperate environments, shoaling behaviour and refuge proximity in contrast may reduce their risk to temperate predators.

Risk-taking behaviour has important consequences for fitness and therefore ecological success of range-extending species [45,46]. Although increased risk-taking can provide benefits such as more food or better habitats, it may also increase mortality risk through increased predator exposure [47]. As such, many animals face a continuous trade-off between predation risk and resource acquisition [48,49]. If resources are limiting (e.g. reduced energy intake), trade-offs may occur in energy allocation towards different processes and behaviours. For example, individuals have to assess risk levels and make decisions to either spend energy fleeing from a threat or preserving energy for other physiological process (*sensu* energy budget theory; [50]). Burst swim responses from predators have a strong energetic cost, which disrupts other fitness-related behaviours such as foraging [51,52]. As such, to maintain physiological homeostasis, continuous decisions are made based on starvation–predation risk trade-offs among behavioural traits (*sensu* economic hypothesis; [53]). However, climatic and biotic alterations can affect the decision-making of organisms and consequently their behavioural responses, which in turn can affect survivorship [54].

Here, we investigate how tropical fishes adjust their antipredator behaviours as they extend their ranges to temperate ecosystems under climate change. These ecosystems at the leading distribution edges of tropical species represent potentially hostile environments, with novel predators and shelter habitats in addition to suboptimal temperatures. We compare various antipredator behaviours between tropical and sympatric native-temperate fish species at multiple locations across a 730 km latitudinal temperature gradient along the southeast Australian coast. To understand if altered antipredator behaviours lead to trade-offs in other behaviours, we also quantified vital behaviours such as foraging, activity level and shelter behaviour, as well as fish body condition, all of which affect species survival and individual fitness [55,56]. Phenotypic flexibility in behaviours by range-extending species in response to novel predators and shelter habitats may be a key determinant of their invasion success at their temperate range locations and therefore a strong regulator of their persistence and expansion in temperate environments (figure 1).

## 2. Material and methods

### (a) Risk-averse behaviours in fishes

Risk-averse behaviours of three tropical species were quantified along a 730 km latitudinal gradient (covering tropical, subtropical and temperate range positions) and compared to that of two temperate species (electronic supplementary material, figure S1). An artificial threat-eliciting stimulus was used to mimic a potential predator attack. The stimulus was created using a cubical PVC frame connected to a 60 cm iron rod, which supported a 30 cm metal ruler at its most distal end (see inset electronic supplementary material, figure S2; following [57]). A GoPro camera was fixed to the cubical frame and positioned towards the ruler. Once a target fish was found, the snorkeler carefully approached the fish and once close enough to the fish the end of the ruler was moved from above the fish towards its head at a constant speed, while the camera was recording its escape behaviour (recording at 30 frames per second). Nagelkerken *et al.* [58] showed that this approach is a good proxy for escape behaviours from real predators. The approach by the snorkeler towards the focal fish always started from approximately the same distance as the water visibility was very similar among the study sites.

Risk-averse behaviours towards a threat were evaluated as a proxy of antipredator performance, using three behavioural proxies (flight initiation distance, escape distance, escape speed)

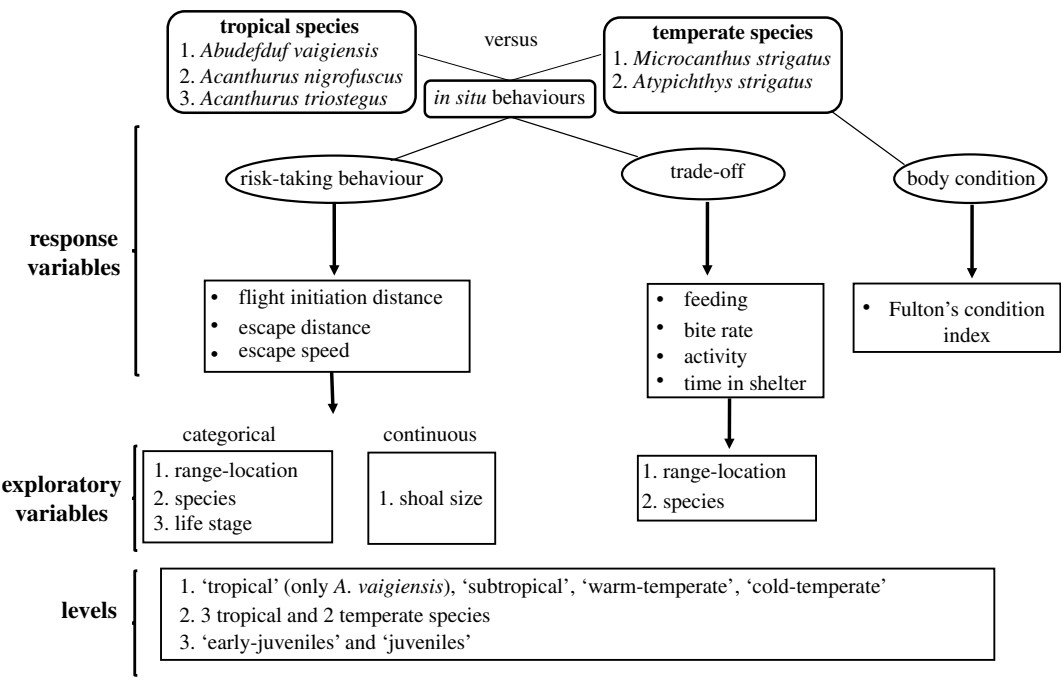

**Figure 1.** Diagram showing the study design. Tropical and temperate fish communities were analysed together for each behaviour and body condition metric using permutational non-parametric ANOVA to test for differences among range locations, fish species and life stage (only for antipredator behaviours).

which were quantified using VLC media player (see details in electronic supplementary material).

## (b) Behaviours modulated by predation risk

We evaluated whether there was a trade-off between risk-assessment behaviours and other important behaviours at multiple sites across a latitudinal gradient (all species). To evaluate this, we analysed feeding behaviour (proportion of time feeding), bite rate (number of bites on the substrate and water column), boldness (time inside shelter) and activity levels (proportion of time actively moving around) for the same focal tropical and temperate fish species. Activity level differs from feeding as in the latter fishes were mostly stationary during feeding. The focal individuals were randomly selected from the same sites where the escape behaviours were measured. However, the behaviours modulated by predation risk and the antipredator behaviours, quantified using two methodologies, were performed at different periods during the day with an interval of a few hours to avoid any behavioural disturbances caused by the artificial fear-eliciting stimulus. For sample sizes see electronic supplementary material, methods.

Underwater GoPro cameras were attached to dive weights and positioned in front of a target fish at a distance of approximately 50 cm. This distance was chosen based on being able to accurately film fishes with a relatively small body size (less than or equal to 5 cm) versus maintaining some distance to avoid disturbing them with the presence of a camera. Each video recording was 10 min long, with the first 3 min used as acclimation time and not included in the analyses.

Behaviours were quantified using the software VLC media player and for each focal individual all behaviours were analysed from the same recording (see details in the electronic supplementary material).

## (c) Body condition

Fulton's condition index was used as a proxy of energy reserve in fish body and growth condition across the latitudinal range locations (electronic supplementary material).

## 3. Results

### (a) *in situ* behaviours across range locations

The tropical and temperate species showed an approximately 1.2- to 1.4-fold increase in flight initiation distance (i.e. higher risk aversion) at the warm-temperate range location compared to the subtropical range location (figure 2a; electronic supplementary material, table S2, $F = 6.888$, $p = 0.033$), with the exception of the temperate species *A. strigatus* whose escape behaviour could not be assessed at the subtropical range location. *A. vaigiensis*, however, showed similar flight initiation distances at the tropical and all other range locations. All the tropical and temperate fishes maintained their escape distances ($F = 0.825$, $p = 0.576$) and escape speeds ($F = 1.134$, $p = 0.424$) across range locations. The escape distance varied from 2 to 26 cm (mean = ∼13 cm) for tropical fishes and 3 to 23 cm (mean = ∼12 cm) for temperate species, and the escape speed ranged from 0.22 to 4.88 cm s$^{-1}$ (mean = 0.9 cm s$^{-1}$) and 0.17 to 2.83 cm s$^{-1}$ (mean = 0.8 cm s$^{-1}$), respectively (electronic supplementary material, figure S3 and table S2). Life stage did not show an effect on the three proxy of antipredator behaviours (flight initiation distance: $F = 1.636$, $p = 0.201$; escape distance: $F = 0.020$, $p = 0.863$; escape speed: $F = 0.086$, $p = 0.801$).

Feeding activity and bite rate did not differ with range location for any of the species (figure 2b,c; electronic supplementary material, table S3, feeding: $F = 0.905$, $p = 0.478$; bite rate: $F = 0.021$, $p = 0.994$). However, activity level showed a decrease (ranging from 18% to 77%) with increasing range location for three of the five species (figure 2d,e; electronic supplementary material, table S3; range location × species, $F = 2.500$, $p = 0.023$), and shelter use increased (ranging from ∼45% to 379%) with range location for two tropical species (range location × species, $F = 2.364$, $p = 0.031$). No difference in the body condition proxy was detected as a function of range location for any of the

Proc. R. Soc. B 289: 20212676

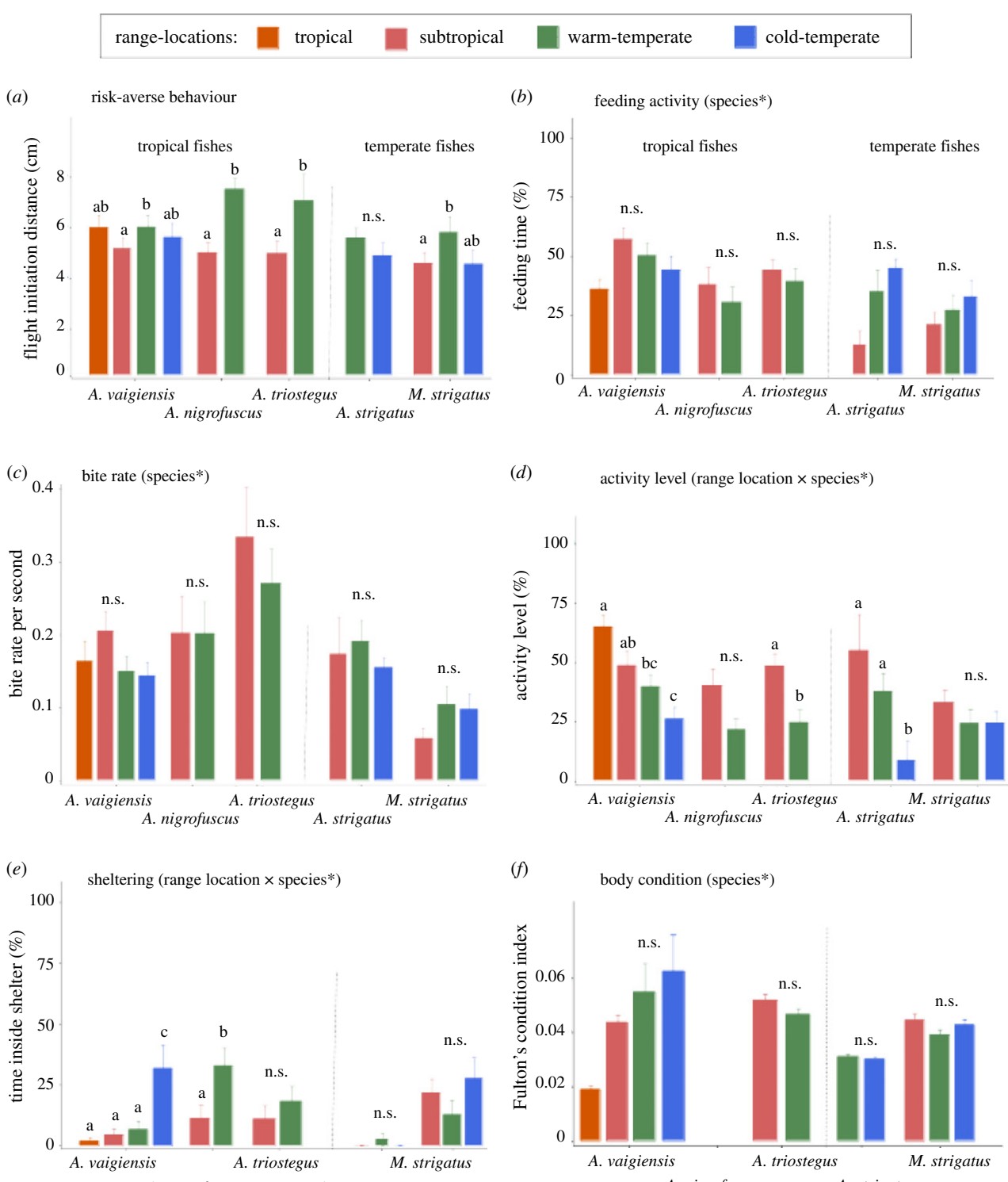

**Figure 2.** *in situ* behaviours (mean + s.e.): (*a*) flight initiation distance as a proxy for risk-averse behaviour towards a threat stimulus, (*b*) proportion of the time feeding, (*c*) bite rate per second, (*d*) proportion of time active, (*e*) proportion of the time sheltering and (*f*) body condition of tropical and temperate species as a function of distributional range locations (tropical only for *A. vaigiensis*, subtropical, warm-temperate, cold-temperate). Different letters above bars indicate significant differences and asterisks indicate when factors such as species or the interaction between range position × species were statistically significant (see electronic supplementary material, tables S2–S4). n.s. = not significant.

species (figure 2*f*; electronic supplementary material, table S3; *F* = 10.162, *p* = 0.088).

## (b) *in situ* behavioural differences of tropical vagrant versus native-temperate fishes

Only the fitness-related behaviours (feeding activity, bite rate, activity level and sheltering) and body condition metric differed significantly among species, and two of these

behaviours differed among species within range location. Two out of three tropical species (*A. triostegus* and *A. vaigiensis*) showed higher proportions of feeding activity than the temperate fishes, independent of the range location (electronic supplementary material, table S3; *F* = 9.375, *p* = 0.0002). Across all studied range locations, all the tropical fishes had higher bite rates than the temperate species *M. strigatus* (species: *F* = 11.349, *p* = 0.0002). The tropical species *A. vaigiensis* showed a higher activity level (range location × species:

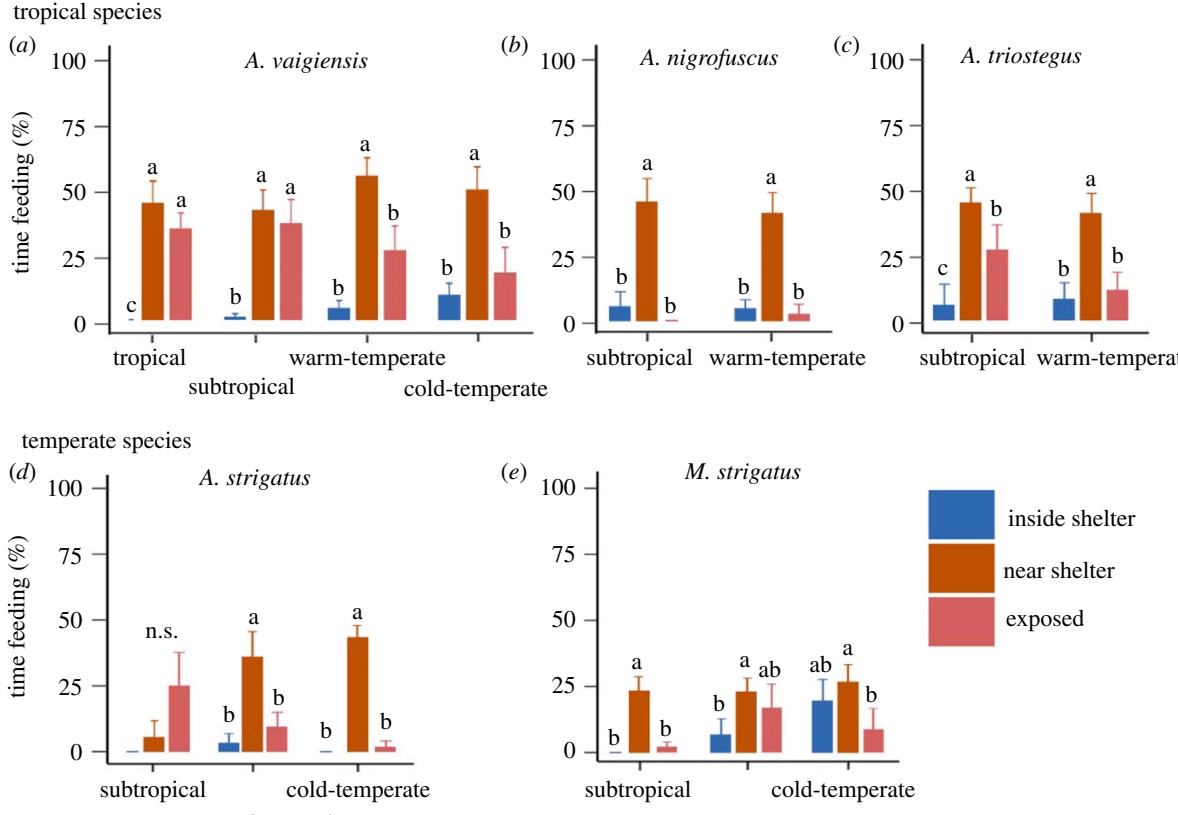

**Figure 3.** The proportion of time (mean + s.e.) that tropical (*a–c*) and temperate (*d,e*) fish species spent feeding when inside shelter, near shelter (distance of 0–5 body lengths from shelter) and away from shelter (exposed; distance greater than five body lengths from shelter) as a function of distributional range location ('tropical' only for *A. vaigiensis*, 'subtropical', 'warm-temperate' and 'cold-temperate'). Different letters above bars indicate significant differences within range locations (see electronic supplementary material, table S5).

$F = 2.500$, $p = 0.023$) than some temperate species at some of the range locations, while the other tropical species *A. nigrofuscus* spent more time inside shelter at its warm-temperate range location than all temperate species (range location × species: $F = 2.364$, $p = 0.031$).

Tropical species had a higher body condition than both of the temperate species, irrespective of range location (electronic supplementary material, table S4; $F = 7.336$, $p = 0.005$).

### (c) Feeding–sheltering trade-offs

Except for a few non-significant trends, tropical and temperate species, in general, showed higher feeding activity in the vicinity of shelter than in exposed areas or while sheltering under rocky overhangs or between crevices (figure 3; electronic supplementary material, tables S5, range location × species × shelter position: $F = 1.828$, $p = 0.043$). Feeding time differed among species as a function of shelter position, with several tropical fishes spending more time feeding in exposed areas and near to shelter than the temperate species, especially at the subtropical range location (species × shelter position, $F = 4.245$, $p = 0.0001$).

### (d) Shoaling group size versus antipredator behaviours

*Abudefduf vaigiensis* and its co-shoaling temperate species *M. strigatus* were the only species that showed a significant, albeit weak effect of shoal size for one of their antipredator behaviours. Escape distance of *A. vaigiensis* decreased with increased shoal size (electronic supplementary material, figure S4; $R^2 = 0.09$, $p = 0.035$), while that for *M. strigatus*

increased (electronic supplementary material, figure S4; $R^2 = 0.13$, $p = 0.036$).

## 4. Discussion

We here show that tropical fishes in general expressed more risk-averse behaviours at their warm-temperate range location than at their subtropical distributional range. Higher risk aversion by coral reef fishes in their novel warm-temperate ranges was expressed as an increased flight initiation distance and/or increased amount of time sheltering in temperate rocky habitats under a perceived threat. Increased risk-averse behaviours enhance the probability of successfully evading or avoiding predatory attacks [59], especially while foraging in unfamiliar habitats [60] or when there is lack of a co-evolutionary prey–predator interaction [46]. The conservative behaviours exhibited by tropical fishes with increasing distance from their native ranges are therefore likely due to uncertainty of the risk factors in an unfamiliar temperate environment [55,61]. In novel habitats, the reliability of the surrounding information on which to judge risk levels is often reduced [55,62]. Thus, exhibiting greater caution is apparently an efficient strategy to reduce predation risk while learning to cope with novel threats [44]. Surprisingly, the tropical species, *A. vaigiensis*, did not show highest flight initiation distance at their most extreme leading range edge (i.e. cold-temperate range location). However, their increased risk-averse behaviour at this range location was shown by the largest time expenditure inside shelters, suggesting that the low-flight initiation distance of *A. vaigiensis* at

cold-temperate range location could be reflected by their higher proximity to shelters. Yet, the relationship between flight initiation distance and distance to shelter is needed to confirm this argument.

Risk-averse behaviours at warm-temperate (all species) and cold-temperate (only for *A. vaigiensis*) range locations did not seem to compromise the fitness of tropical range-extending fishes. We show that an increased flight response at temperate locations was not associated with reduced foraging (e.g. feeding activity or bite rates) or reduced body condition. However, this finding can be related to the fact that individuals that under-responded to temperate predators (i.e. allowed closer approach) were less observed in this study because they were already consumed by predators. Although animals that over-respond to a threat often engage less in other fitness-related activities [63,64], some animals compensate this by adjusting the time allocated to other behaviours, i.e. known as the predation risk compensation term [56,65,66]. Accordingly, two out of three tropical fish species showed increased sheltering behaviours at temperate locations which was associated with highly reduced feeding (i.e. when inside shelter). Hence, the reduced activity levels at temperate locations may be a behavioural strategy adopted by some tropical fishes to preserve energy in the light of this reduced feeding, and therefore maintain their body condition, albeit at the cost of reduced growth [67]. The water temperatures during the field surveys at the tropical locations in autumn and winter were comparable to those from the subtropical and warm-temperate locations sampled during summer. This supports the notion that any behavioural differences especially for *A. vaigiensis* among locations were primarily driven by response to a novel environment rather than to seawater temperature *per se*. This suggests that tropical fishes trade-off activity levels (through increased time sheltering) for maintenance of fitness as an adaptive response to their novel environment. Such trade-offs can ensure the future survivorship of some tropical fishes at temperate locations as winter temperatures continue to increase due to climate change. Hence, the balance between risk-avoidance, activity levels and feeding behaviours can be critical for tropical range-extending species at the initial phase of their range extension. However, eventually this could diminish as tropical vagrants learn to recognize temperate predators and behaviourally adjust to their novel surrounding environment.

Similar to tropical fishes, one of the studied temperate species (*M. strigatus*) was more risk-averse at the warm-temperate range position, while the other temperate species (*A. strigatus*) was less active at the cold-temperate range position. Hence, our prediction that native-temperate fishes would maintain their behavioural responses across distributional ranges, because these species have always naturally occurred there, was not supported by our findings. At the same time, our study reveals that native-temperate fishes did not show a consistently altered pattern in their behaviours across their range locations, as one species increased its flight initiation distance at the warm-temperate location and the second species decreased its activity level at the coldest range location.

Tropical range-extending fishes had higher fitness-related behaviours (i.e. higher overall feeding activity and bite rates, and in some cases also higher activity levels and less time inside shelter) and higher body condition than their co-shoaling native-temperate species. These results may be surprising because a logical prediction would be that temperate fishes perform better in their native-temperate ranges than tropical vagrants. However, higher foraging performance of tropical fishes compared to temperate fishes could be a natural physiological response to meet their specific metabolic demands, or a more general response to unfamiliar and stressful environments. Even though temperate ecosystems are novel environments for tropical fishes, shoaling with temperate fish species can affect their performance both positively and negatively. For example, shoaling with temperates may benefit tropical fishes by attaining larger body sizes compared to 'tropical-only' shoals [38]. By contrast, in the presence of artificially supplemented food (i.e. experimentally enforced competition), temperate fishes show increased feeding performance and higher aggression towards tropical fishes, and tropicals show increased sheltering with higher abundances of temperate fishes [68]. However, such competitive effects might be mitigated under more natural conditions, through behavioural [69] and trophic [70] niche segregation between tropical and temperate fishes. In addition, ocean warming and acidification are likely to diminish shoaling performance of mixed tropical-temperate fish shoals, diminishing any gained effects (if any) of mixed shoaling [12]. Hence, under future climate conditions, the performance of tropical range-extending fishes will not only depend on shoaling dynamics and niche segregation as shown previously, but also on behaviourally mediated adjustments to feeding and risk aversion.

## 5. Conclusion

We reveal that tropical fishes can show more risk-averse behaviours at their warm and cold-temperate distributional ranges under ongoing climate change (i.e. increased flight initiation distance and time in shelter) without affecting their body condition. The way in which tropical fish behaviourally balance the conflicting demands between avoiding temperate predators and sustaining feeding, combined with decreased activity levels and increased sheltering, is critical for maintaining their fitness and can have flow-on effects on temperate fish communities under ocean warming.

**Ethics.** All experiments were performed under animal ethics approval nos. S-2015-222A and S-2017-002 (University of Adelaide) and ETH17-1117 (University of Technology Sydney) and followed the University's animal ethics guidelines.

**Data accessibility.** Data are available from the Dryad Digital Repository: https://doi.org/10.5061/dryad.gmsbcc2qb [71].

**Authors' contributions.** E.O.C.C.: conceptualization, data curation, formal analysis, investigation, methodology, writing—original draft and writing—review and editing; D.J.B.: conceptualization, methodology, supervision and writing—review and editing; I.N.: conceptualization, funding acquisition, investigation, methodology, project administration, supervision and writing—review and editing.

All authors gave final approval for publication and agreed to be held accountable for the work performed therein.

**Competing interests.** The authors declare no competing interests.

**Funding.** Financial support was provided by a Discovery Projects grant from the Australian Research Council to I.N. and D.J.B. (grant no. DP170101722).

**Acknowledgements.** We thank Minami Sasaki, Kelsey Kingsbury, Camilo Ferreira, Angus Mitchell, Chloe Hayes, Michaela Krutz, Mitchell Brennan and Renne Ashton for logistic support in the field.

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
