## [Peer Review File · Proceedings of the Royal Society B: Biological Sciences]

Review History

RSPB-2021-2676.R0 (Original submission)

Review form: Reviewer 1

Recommendation

Accept with minor revision (please list in comments)

Scientific importance: Is the manuscript an original and important contribution to its field?

Good

General interest: Is the paper of sufficient general interest?

Acceptable

Quality of the paper: Is the overall quality of the paper suitable?

Good

Is the length of the paper justified?

Yes

Should the paper be seen by a specialist statistical reviewer?

No

Do you have any concerns about statistical analyses in this paper? If so, please specify them explicitly in your report.

No

It is a condition of publication that authors make their supporting data, code and materials available - either as supplementary material or hosted in an external repository. Please rate, if applicable, the supporting data on the following criteria.

Is it accessible?

Yes

Is it clear?

N/A

Is it adequate?

N/A

Do you have any ethical concerns with this paper?

No

Comments to the Author

I am pleased to see that the suggestions and comments of the reviewers have been comprehensively acted upon. The vast majority of the responses I feel covered my concerns and have improved the manuscript substantially. Most importantly, there is addition of tropical sites, the removal of the seasonal comparison and the comparison between tropical and temperate species. While the patterns in behaviour revealed by this extra work are not always clear-cut, I believe these analyses have made the paper much stronger.

Thank you for clarifying how activity and foraging were assessed. It has ameliorated my concerns somewhat. I still worry a little about activity levels since it reads to me as if position of the fish were taken at the start of the two intervals, and distance travelled assessed from these. Thus, activity could be being underestimated, or missed. However, this is a relatively minor concern, and I believe that any underestimate would be consistent across the study.

Despite the laudable extra effort, my primary issue with the paper is that I believe the authors are, in their headline statements in the title, abstract and to a lesser extent in the conclusions, drawing too much certainty from evidence that while indicative, is most certainly not conclusive. For example, their title "Coral-reef fishes are more risk-averse at their poleward range limits" elides the fact that temperate species also show higher risk aversion towards poleward limits (e.g. for FID in *M. strigatus* and activity levels in *A. strigatus*), while the tropical species *A. viagiensis* and *A. triostegus* do not show more wary behaviours at the furthest poleward site (FID and time inside shelter respectively).

In my opinion, because latitude is treated as a fixed factor rather than a continuous variable, the testing is for differences between locations, rather than latitude per se. I think the use of 'range location' as has been done in general here is correct, but especially in some of the supplementary information and the abstract the use of "latitude" and "latitudinal temperature gradient" give the impression of firmer patterns than shown here.

I also worry about consistency in the interpretation of the patterns that were found. For example, in the abstract you write "Tropical fishes became more risk-averse (increased shelter use and flight initiation distance) and their activity level decreased poleward ... native temperate fishes did not show a consistently altered pattern in their behaviours across latitude." There was not a consistent increase in risk-aversion within the tropical species (*A. viagiensis* had lower FID at the most poleward site, *A. nigrofuscus* showed no difference in activity levels while the other two species did, and *A. triostegus* showed no increased time inside shelter. Considering that there was evidence of increased risk aversion for *A. strigatus* (activity level declined) and *M. strigatus* showed similar FID changes to *A. viagiensis*, I think that it could be argued that the tropical species also did not show a consistently altered pattern in behaviours across latitude.

Regarding the FID, it appears there may be an effect of the warm-temperate location, with FID returning to equivalent levels as the subtropical locations for both the tropical and temperate species that were measured at both cold-temperate and subtropical. For that matter, for *A. viagiensis*, the FID at the tropical sites were similar to the warm-temperate location, which may suggest that the FID measured at the subtropical site were anomalously low. While you suggest that decreased FID for *A. viagiensis* at the cold-temperate site may be due to the adoption of other anti-predator strategies (more time in shelter), *M. strigatus* which shows similar reduced FID to *A. viagiensis* does not adopt this behaviour. Regardless of this, I believe your argument for this is not correct, because while spending more time in shelter suggests that fish have adopted extra anti-predator measures, it does not then follow that FID would be reduced. If anything, I would suggest that increased wariness measured by shelter use should be reflected in higher FID when the fish is away from shelter. If the fishes were moving less far from shelter, this may explain lower FID, but would require an analysis of FID vs distance from shelter to confirm. Finally, considering that there was no difference found in escape distances and escape speeds (line 329-330), suggesting as the authors do on lines 395 -397 that there was an increased flight response at higher latitudes, is not correct across all tropical species, and as mentioned, a similar response was found for at least one temperate species.

The evidence is stronger for latitude having an impact on activity levels, but again there is evidence of this for both tropical and temperate species, with one species of each group showing no effect of latitude. The first line of the Discussion, that “We here show that tropical fishes expressed more risk-averse behaviours at their warm-temperate range location than at their subtropical distributional range,” is accurate, but I think that the statement in the conclusion that ‘tropical fishes can show increased risk aversion at the centre of their temperate distributional ranges’ at line 439 - 440 is reaching for a latitudinal explanation which the evidence doesn’t support. If more sites around this warm-temperate location were studied with similar results, that would provide this evidence. At the moment I fear it is a location effect not a latitudinal effect.

The comparisons between tropical and temperate species were useful, and suggest that there are species differences, with much of this difference being driven by *A. viagiensis* which as mentioned in an earlier review, is quite different from either of the other tropical species. However, latitude does not seem to be having an impact on shelter position, body condition, feeding or bite rate. I stress that I think that their underlying hypothesis may be sound, but that the evidence presented at best only weakly support it.

A more minor critique is that in the introduction there is little discussion of neophobia, which has been heavily studied in fishes and would be relevant to any discussion here. There are multiple studies on risk-induced neophobia in fishes (Joyce et al. 2016; Chivers et al. 2014) In particular, the work on neophobia has suggested that larval fish exposed to high-risk stimuli (such as olfactory cues) may show heightened responses to those or similar cues in later trials. Overall, I do believe that there is merit to the concept that life-stage is an important determinant of predation risk, but this discussion of the influence of life-stage on susceptibility to predation should address neophobia, particularly because of the overall argument that tropicalising species will be introduced to novel environments and predators and the ability to respond would be important for establishment. Indeed, the authors cite several studies on neophobia, which begs the question why this is not addressed directly in the introduction (to be clear, I do not think this should make significant substantial changes in predictions, but it is a missing part of the literature)

Other minor comments

Line 56-57: Think about rewording this slightly – it gave the impression on first reading that the entire species was relieved from historically important predators, not the part of the species/individuals that were expanding their range.

Line 76: replace ‘can’ with ‘could’ – temperature reducing specifically ‘anti-predator performance’ has not been demonstrated in the work cited. Or provide citation.

Line 80 – 82: I suggest removing this line because it is an oversimplified statement of the influence of life stage for predation susceptibility in fishes.

Line 94-96: I don't think it has been demonstrated that early stages of coral reef fishes are more vulnerable to predation due to unfamiliarity to temperate climates and habitats – this has been argued in the introduction, but not demonstrated.

Figure 2 – can you state what the asterisked labels mean in the figure.

Chivers DP, McCormick MI, Mitchell MD, Ramasamy RA, Ferrari MC. Background level of risk determines how prey categorize predators and non-predators. *Proceedings of the Royal Society B: Biological Sciences*. 2014 Jul 22;281(1787):20140355.

Joyce BJ, Demers EE, Chivers DP, Ferrari MC, Brown GE. Risk-induced neophobia is constrained by ontogeny in juvenile convict cichlids. *Animal Behaviour*. 2016 Apr 1;114:37-43.

Samia DS, Blumstein DT, Stankowich T, Cooper Jr WE. Fifty years of chasing lizards: new insights advance optimal escape theory. *Biological Reviews*. 2016 May;91(2):349-66.

Review form: Reviewer 2

Recommendation

Accept with minor revision (please list in comments)

Scientific importance: Is the manuscript an original and important contribution to its field?

Good

General interest: Is the paper of sufficient general interest?

Good

Quality of the paper: Is the overall quality of the paper suitable?

Good

Is the length of the paper justified?

Yes

Should the paper be seen by a specialist statistical reviewer?

No

Do you have any concerns about statistical analyses in this paper? If so, please specify them explicitly in your report.

No

It is a condition of publication that authors make their supporting data, code and materials available - either as supplementary material or hosted in an external repository. Please rate, if applicable, the supporting data on the following criteria.

Is it accessible?

N/A

Is it clear?

N/A

Is it adequate?

N/A

Do you have any ethical concerns with this paper?

No

Comments to the Author

The manuscript by Coni and colleagues presents a challenging and integrating in situ experiment to try and understand the real world outcomes for range shifting species when entering a novel system. My first impression of the manuscript is that it takes the work generally done by this team to the next level to attempt to understand the ecological processes for marine vagrant fish. It is a complex experiment that the authors have done a good job to try and distil to a clear story.

There are some interesting patterns in the results presented that indicate increased risk-aversion. I would love to see more data from the same locations with a greater range of water temperatures as I think it would elucidate what is due to location vs temperature. Just a few ideas and comments from me

I am passionate about a good study design in the methods sections, and due to the complexity of the sampling that occurred in the manuscript I think it forms an essential component. In Fig 1, is there not supposed to be a line going to the trade-off round circle, I assume from "in situ behaviours"?

Should there also be a number(s) in front of shoal size for the continuous explanatory variable?

I appreciate the addition of the tropical locations for *A. vaigiensis*. It fascinated me that in the water temperatures for Heron and Magnetic Island were quite close to the sub-tropical and warm-temperate locations (Table S1). So if a response was largely driven by physiological processes linked to temperature, e.g. metabolic rate, then it could explain/support why a difference is not seen between the groups. For example feeding. Do the authors think that this similarity in temperature might be driving the patterns with feeding they saw? I think experimentally the Booth lab would have an idea how temperature changes feeding rate.

In cases where a difference is seen, like activity level the responses found here can't be temperature driven, instead that seems to be likely due to the difference in habitat/ecosystem, right?

The pattern of body condition getting higher (though surprisingly NS) for *A. vaigiensis* is interesting. Was this an artefact perhaps of SL reducing with cooler locations?

I had trouble working out which fish were collected and body condition measured in and what the replication was. I can see that the same fish were used in both observational testing procedures, some did not give good data for both of the observations I assume. So then I am just not sure how the morphology matches up.

On this if all fish that were observed were collected, why not have size as a continuous covariate, rather than blocked?

The link the authors make between risk-aversion and fitness is fine, but it seems that a caveat line should be there at least to admit that these might be the fish that are left, all the other foolish ones may have been eaten. And in the case of the FK condition it may be that there is an optimum that is being selected for.

Line 173: This is the first time the convict surgeon species name is mentioned, and since all tropical species genus starts with an "A", I would give genus in full for specificity.

Line 261: I don't think sec and min need "." after

Line 239-333: Would have been nice to have some values of what the escape speeds and distances were within the main text without having to go look at the Supl. Just a basic range in the brackets where the stats are would be good.

Fig 2, I would have preferred the legend at the top. And maybe something to make the text that indicates the model significant more distinct would be good.

At first I was surprised that the Supl Figures not have the same colour scheme. But then I realised this was an amendment in the revision. I would change Supl figures so that they match main text figures and the map?

Line 417: Logical rather than logic

Decision letter (RSPB-2021-2676.R0)

24-Jan-2022

Dear Dr Nagelkerken:

Your manuscript has now been peer reviewed and the reviews have been assessed by an Associate Editor. The reviewers' comments (not including confidential comments to the Editor) and the comments from the Associate Editor are included at the end of this email for your reference. As you will see, the reviewers and the Editors have raised some concerns with your manuscript and we would like to invite you to revise your manuscript to address them. While overall, there is a consensus that your manuscript contains some valuable and informative findings, I share the concern about the potential for overstatement in some of the findings. It is important, within the usual restrictions imposed by resources that many experimental and sampling designs have caveats associated with them. I would ask you to pay particular attention to these elements in your revised manuscript. As you will see below, the invitation to revise, does not guarantee eventual publication, and it would be most helpful for the onward peer-review, to provide a sufficiently informative and constructive response letter and annotated modifications to the manuscript.

Research ethics:

Use of animals and field studies:

It is a condition of publication that you make available the data and research materials supporting the results in the article. Please see our Data Sharing Policies (<https://royalsociety.org/journals/authors/author-guidelines/#data>). Datasets should be deposited in an appropriate publicly available repository and details of the associated accession number, link or DOI to the datasets must be included in the Data Accessibility section of the article (<https://royalsociety.org/journals/ethics-policies/data-sharing-mining/>). Reference(s) to datasets should also be included in the reference list of the article with DOIs (where available).

Please submit a copy of your revised paper within three weeks. If we do not hear from you within this time your manuscript will be rejected. If you are unable to meet this deadline please let us know as soon as possible, as we may be able to grant a short extension.

Best wishes,
Professor Gary Carvalho
<mailto:proceedingsb@royalsociety.org>

Associate Editor

Comments to Author:

This is the second submission of this manuscript and it has an unusual case at PRSB since the first Contributing Editor was unable to handle it a second time. It was the right thing to do to pass it to another (me), and I am pleased that we have had it reviewed, especially since one review was from the same person that evaluated the first submission. As this topic is not terribly close to my research expertise, I have to heavily rely on the opinion of the reviewers.

The manuscript addresses range extension in tropical fishes, and the possibility that risk aversion in novel habitats might mediate their capacity to extend their ranges in a changing world. As noted by the reviewers, these kinds of studies are highly challenging to conduct, and it is impressive that the authors have brought this off at all. The fact that the study has limitations is, therefore, not surprising, and they should not constitute a reason to reject the submission. Both reviewers are supportive of the submission, although they consistently highlight the view that the authors are reaching a little too far with their inferences and summary statements. Reviewer #1 is the most critical in this regard, and they raise the possibility that this study should be marketed as a contrast among sites rather than a sampling across a latitudinal gradient. I do not think this is a critical issue, as the sampling has been conducted at multiple sites across a remarkable range of latitudes.

The topic of this submission remains suitable for PRSB and it should have wide appeal to the readership. The quality of the work meets the standard required for the journal, and I think it would make a fine contribution. Given the consistently expressed view that the authors are reaching a little too far with their concluding statements, it would be appropriate to request minor changes to better match the summary statements to the scope of the data.

Board Member: 2

Comments to Author(s):

This is an interesting manuscript that has been re-submitted by Coni et al. in response to an initial submission and recommendation/invitation by a PRSB Board Member. As this person is no longer available, the manuscript has landed on my desk even though the subject material is a little removed from my area of expertise. I have strong feelings about the need to honor the previous arrangement, particularly as the authors have made substantial changes in response to the first set of review comments. Although the window of resubmission has closed under the previous arrangement, I am suggesting this should be returned to the same reviewers and ask them whether their concerns have been addressed.

The current form of the manuscript remains appropriate for PRSB and the content has wide appeal to the readership of the journal. In brief, it addresses the role of behavior in mediating range extension in fishes, and uses tropical and temperate examples as model systems. The manuscript reports the fascinating result that tropical fishes become more risk averse in locations more poleward of their tropical distributions, potentially facilitating their range extensions under climate change conditions.

Reviewer(s)' Comments to Author:

Referee: 1

Comments to the Author(s)

I am pleased to see that the suggestions and comments of the reviewers have been comprehensively acted upon. The vast majority of the responses I feel covered my concerns and have improved the manuscript substantially. Most importantly, there is addition of tropical sites, the removal of the seasonal comparison and the comparison between tropical and temperate species. While the patterns in behaviour revealed by this extra work are not always clear-cut, I believe these analyses have made the paper much stronger.

Thank you for clarifying how activity and foraging were assessed. It has ameliorated my concerns somewhat. I still worry a little about activity levels since it reads to me as if position of the fish were taken at the start of the two intervals, and distance travelled assessed from these. Thus, activity could be being underestimated, or missed. However, this is a relatively minor concern, and I believe that any underestimate would be consistent across the study.

Despite the laudable extra effort, my primary issue with the paper is that I believe the authors are, in their headline statements in the title, abstract and to a lesser extent in the conclusions, drawing too much certainty from evidence that while indicative, is most certainly not conclusive. For example, their title "Coral-reef fishes are more risk-averse at their poleward range limits"

elides the fact that temperate species also show higher risk aversion towards poleward limits (e.g. for FID in *M strigatus* and activity levels in *A strigatus*), while the tropical species *A viagiensis* and *A triostegus*. do not show more wary behaviours at the furthest poleward site (FID and time inside shelter respectively).

In my opinion, because latitude is treated as a fixed factor rather than a continuous variable, the testing is for differences between locations, rather than latitude per se. I think the use of 'range location' as has been done in general here is correct, but especially in some of the supplementary information and the abstract the use of "latitude" and "latitudinal temperature gradient" give the impression of firmer patterns than shown here.

I also worry about consistency in the interpretation of the patterns that were found. For example, in the abstract you write "Tropical fishes became more risk-averse (increased shelter use and flight initiation distance) and their activity level decreased poleward ... native temperate fishes did not show a consistently altered pattern in their behaviours across latitude." There was not a consistent increase in risk-aversion within the tropical species (*A viagiensis* had lower FID at the most poleward site, *A nigrofusus* showed no difference in activity levels while the other two species did, and *A triostegus* showed no increased time inside shelter. Considering that there was evidence of increased risk aversion for *A strigatus* (activity level declined) and *M strigatus* showed similar FID changes to *A viagiensis*, I think that it could be argued that the tropical species also did not show a consistently altered pattern in behaviours across latitude.

Regarding the FID, it appears there may be an effect of the warm-temperate location, with FID returning to equivalent levels as the subtropical locations for both the tropical and temperate species that were measured at both cold-temperate and subtropical. For that matter, for *A viagiensis*, the FID at the tropical sites were similar to the warm-temperate location, which may suggest that the FID measured at the subtropical site were anomalously low. While you suggest that decreased FID for *A viagiensis* at the cold-temperate site may be due to the adoption of other anti-predator strategies (more time in shelter), *M strigatus* which shows similar reduced FID to *A viagiensis* does not adopt this behaviour. Regardless of this, I believe your argument for this is not correct, because while spending more time in shelter suggests that fish have adopted extra anti-predator measures, it does not then follow that FID would be reduced. If anything, I would suggest that increased wariness measured by shelter use should be reflected in higher FID when the fish is away from shelter. If the fishes were moving less far from shelter, this may explain lower FID, but would require an analysis of FID vs distance from shelter to confirm. Finally, considering that there was no difference found in escape distances and escape speeds (line 329-330), suggesting as the authors do on lines 395 -397 that there was an increased flight response at higher latitudes, is not correct across all tropical species, and as mentioned, a similar response was found for at least one temperate species.

The evidence is stronger for latitude having an impact on activity levels, but again there is evidence of this for both tropical and temperate species, with one species of each group showing no effect of latitude. The first line of the Discussion, that "We here show that tropical fishes expressed more risk-averse behaviours at their warm-temperate range location than at their subtropical distributional range," is accurate, but I think that the statement in the conclusion that 'tropical fishes can show increased risk aversion at the centre of their temperate distributional ranges' at line 439 - 440 is reaching for a latitudinal explanation which the evidence doesn't support. If more sites around this warm-temperate location were studied with similar results, that would provide this evidence. At the moment I fear it is a location effect not a latitudinal effect.

The comparisons between tropical and temperate species were useful, and suggest that there are species differences, with much of this difference being driven by *A. viagiensis* which as mentioned in an earlier review, is quite different from either of the other tropical species. However, latitude does not seem to be having an impact on shelter position, body condition, feeding or bite rate. I stress that I think that their underlying hypothesis may be sound, but that the evidence presented at best only weakly support it.

A more minor critique is that in the introduction there is little discussion of neophobia, which has been heavily studied in fishes and would be relevant to any discussion here. There are multiple studies on risk-induced neophobia in fishes (Joyce et al. 2016; Chivers et al. 2014) In particular, the work on neophobia has suggested that larval fish exposed to high-risk stimuli (such as olfactory cues) may show heightened responses to those or similar cues in later trials. Overall, I do believe that there is merit to the concept that life-stage is an important determinant of predation risk, but this discussion of the influence of life-stage on susceptibility to predation should address neophobia, particularly because of the overall argument that tropicalising species will be introduced to novel environments and predators and the ability to respond would be important for establishment. Indeed, the authors cite several studies on neophobia, which begs the question why this is not addressed directly in the introduction (to be clear, I do not think this should make significant substantial changes in predictions, but it is a missing part of the literature)

Other minor comments

Line 56-57: Think about rewording this slightly – it gave the impression on first reading that the entire species was relieved from historically important predators, not the part of the species/individuals that were expanding their range.

Line 76: replace 'can' with 'could' – temperature reducing specifically 'anti-predator performance' has not been demonstrated in the work cited. Or provide citation.

Line 80 – 82: I suggest removing this line because it is an oversimplified statement of the influence of life stage for predation susceptibility in fishes.

Line 94-96: I don't think it has been demonstrated that early stages of coral reef fishes are more vulnerable to predation due to unfamiliarity to temperate climes and habitats – this has been argued in the introduction, but not demonstrated.

Figure 2 – can you state what the asterisked labels mean in the figure.

Chivers DP, McCormick MI, Mitchell MD, Ramasamy RA, Ferrari MC. Background level of risk determines how prey categorize predators and non-predators. *Proceedings of the Royal Society B: Biological Sciences*. 2014 Jul 22;281(1787):20140355.

Joyce BJ, Demers EE, Chivers DP, Ferrari MC, Brown GE. Risk-induced neophobia is constrained by ontogeny in juvenile convict cichlids. *Animal Behaviour*. 2016 Apr 1;114:37-43.

Samia DS, Blumstein DT, Stankowich T, Cooper Jr WE. Fifty years of chasing lizards: new insights advance optimal escape theory. *Biological Reviews*. 2016 May;91(2):349-66.

Referee: 2

Comments to the Author(s)

The manuscript by Coni and colleagues presents a challenging and integrating in situ experiment to try and understand the real world outcomes for range shifting species when entering a novel system. My first impression of the manuscript is that it takes the work generally done by this team to the next level to attempt to understand the ecological processes for marine vagrant fish. It is complex experiment that the authors have done a good job to try and distil to a clear story. There are some interesting patterns in the results presented that indicate increase risk-aversion. I would love to see more data from the same locations with a greater range of water temperatures as I think it would elucidate what is due to location vs temperature. Just a few ideas and comments from me

I am passionate about a good study design in methods sections, and due to the complexity of the sampling that occurred in the manuscript I think it forms an essential component. In Fig 1 is there not supposed to be line going to the trade-off round circle, I assume from "in situ behaviours"? Should there also be a number(s) in front of shoal size for the continuous explanatory variable? I appreciate the addition of the tropical locations for *A. vaigiensis*. It fascinated me that in the water temperatures for Heron and Magnetic Island were quite close to the sub-tropical and

warm-temperate locations (Table S1). So if a response was largely driven by physiological processes linked to temperature, e.g. metabolic rate, then it could explain/support why a difference is not seen between the groups. For example feeding. Do the authors think that this similarity in temperature might be driving the patterns with feeding they saw? I think experimentally the Booth lab would have an idea how temperature changes feeding rate. In cases where a difference is seen, like activity level the responds found here can't be temperature driven, instead that seems to be likely due to the difference in habitat/ecosystem, right?

The pattern of body condition getting higher (though surprisingly NS) for *A. vaigiensis* is interesting. Was this an artefact perhaps of SL reducing with cooler locations?

I had trouble working out which fish were collected and body condition measured in and what the replication was. I can see that the same fish were used in both observational testing procedures, some did not give good data for both of the observations I assume. So then I am just not sure how the morphology matches up.

On this if all fish that were observed were collected, why not have size as a continuous covariate, rather than blocked?

The link the authors make between risk-aversion and fitness is fine, but it seems that a caveat line should be there at least to admit that these might be the fish that are left, all the other foolish ones may have been eaten. And in the case of the FK condition it may be that there is an optimum that is being selected for.

Line 173: This is the first time the convict surgeon species name is mentioned, and since all tropical species genus starts with an "A", I would give genus in full for specificity.

Line 261: I don't think sec and min need "." after

Line 239-333: Would have been nice to have some values of what the escape speeds and distances were within the main text without having to go look at the Supl. Just a basic range in the brackets where the stats are would be good.

Fig 2, I would have preferred the legend at the top. And maybe something to make the text that indicates the model significant more distinct would be good.

At first I was surprised that the Supl Figures not have the same colure scheme. But then I realised this was an amendment in the revision. I would change Supl figures so that they matches main text figures and the map?

Line 417: Logical rather than logic

Author's Response to Decision Letter for (RSPB-2021-2676.R0)

See Appendix A.

Decision letter (RSPB-2021-2676.R1)

18-Feb-2022

Dear Dr Nagelkerken

I am pleased to inform you that your manuscript entitled "Coral-reef fishes can become more risk-averse at their poleward range limits" has been accepted for publication in Proceedings B.

Data Accessibility section

Open Access

Paper charges

Sincerely,

Professor Gary Carvalho

Associate Editor:

Comments to Author:

The changes in this version have provided a more nuanced and balanced summary of the findings, and the inferences are better matched to the scope of the data and the constraints of the experimental design. This is not a very nice contribution that is appropriate for publication in PRSB; it's appeal is enhanced by the shortening through transfer of some of the methods to ESM.

Appendix A

MS RSPB-2020-1709 Author Response to Comments by Editors and Reviewers

Associate Editor:

Comments to Author:

Dear Dr Nagelkerken and colleagues,

This is the second submission of this manuscript and it has an unusual case at PRSB since the first Contributing Editor was unable to handle it a second time. It was the right thing to do to pass it to another (me), and I am pleased that we have had it reviewed, especially since one review was from the same person that evaluated the first submission. As this topic is not terribly close to my research expertise, I have to heavily rely on the opinion of the reviewers.

The manuscript addresses range extension in tropical fishes, and the possibility that risk aversion in novel habitats might mediate their capacity to extend their ranges in a changing world. As noted by the reviewers, these kinds of studies are highly challenging to conduct, and it is impressive that the authors have brought this off at all. The fact that the study has limitations is, therefore, not surprising, and they should not constitute a reason to reject the submission. Both reviewers are supportive of the submission, although they consistently highlight the view that the authors are reaching a little too far with their inferences and summary statements. Reviewer #1 is the most critical in this regard, and they raise the possibility that this study should be marketed as a contrast among sites rather than a sampling across a latitudinal gradient. I do not think this is a critical issue, as the sampling has been conducted at multiple sites across a remarkable range of latitudes.

The topic of this submission remains suitable for PRSB and it should have wide appeal to the readership. The quality of the work meets the standard required for the journal, and I think it would make a fine contribution. Given the consistently expressed view that the authors are reaching a little too far with their concluding statements, it would be appropriate to request minor changes to better match the summary statements to the scope of the data.

RESPONSE: We have addressed all suggestions and comments of the two reviewers as mentioned below. The principal concern raised was that we reached a little too far with our inferences and summary statements. Hence, we rewrote statements which contained strong inferences, and toned them down so they associated better with our empirical findings.

Board Member: 2

Comments to Author(s):

This is an interesting manuscript that has been re-submitted by Coni et al. in response to an initial submission and recommendation/invitation by a PRSB Board Member. As this person is no longer available, the manuscript has landed on my desk even though the subject material is a little removed from my area of expertise. I have strong feelings about the need to honor the previous arrangement, particularly as the authors have made substantial changes in response to the first set of review comments. Although the window of resubmission has closed under the previous arrangement, I am suggesting this should be returned to the same reviewers and ask them whether their concerns have been addressed.

The current form of the manuscript remains appropriate for PRSB and the content has wide appeal to the readership of the journal. In brief, it addresses the role of behavior in mediating range extension in fishes, and uses tropical and temperate examples as model systems. The manuscript reports the fascinating result that tropical fishes become more risk averse in locations more poleward of their tropical distributions, potentially facilitating their range extensions under climate change conditions.

RESPONSE: We really appreciate the detail and insightfulness of the reviews. We have addressed all suggestions and comments of the two reviewers, including the two major concerns highlighted above by the Associate Editor: (1) we rewrote all statements in the manuscript that were worded too strongly, and (2) the behavioural comparisons are now contrasted among range-locations rather than across a latitudinal gradient.

We have also moved parts of the Methods section to the Supplementary methods due to the page limits of Proc. Roy. Soc B.

Reviewer(s)' Comments to Author:

Referee 1:

I am pleased to see that the suggestions and comments of the reviewers have been comprehensively acted upon. The vast majority of the responses I feel covered my concerns and have improved the manuscript substantially. Most importantly, there is addition of tropical sites, the removal of the seasonal comparison and the comparison between tropical and temperate species. While the patterns in behaviour revealed by this extra work are not always clear-cut, I believe these analyses have made the paper much stronger.

Thank you for clarifying how activity and foraging were assessed. It has ameliorated my concerns somewhat. I still worry a little about activity levels since it reads to me as if position of the fish were taken at the start of the two intervals, and distance travelled assessed from these. Thus, activity could be being underestimated, or missed. However, this is a relatively minor concern, and I believe that any underestimate would be consistent across the study.

RESPONSE: Indeed, our video recordings did not allow us to measure the true distances traveled by the fish. Instead, we measured the proportion of time a fish was active by quantifying during how many intervals they changed their position. As suggested by the reviewer we added in the methods (supplementary information – ‘Behaviour modulated by predation risk’) a statement about the consistency of the approach: “As the activity level was measured in the same way (start of the intervals) at all sites, the under- or overestimation that could occur are consistent across the study sites.”

Despite the laudable extra effort, my primary issue with the paper is that I believe the authors are, in their headline statements in the title, abstract and to a lesser extent in the conclusions, drawing too much certainty from evidence that while indicative, is most certainly not conclusive. For example, their title “Coral-reef fishes are more risk-averse at their poleward range limits” elides the fact that temperate species also show higher risk aversion towards poleward limits (e.g. for FID in *M. strigatus* and activity levels in *A. strigatus*), while the tropical species *A. viagiensis* and *A. triostegus*. do not show more wary behaviours at the furthest poleward site (FID and time inside shelter respectively).

In my opinion, because latitude is treated as a fixed factor rather than a continuous variable, the testing is for differences between locations, rather than latitude per se. I think the use of ‘range location’ as has been done in general here is correct, but especially in some of the

supplementary information and the abstract the use of “latitude” and “latitudinal temperature gradient” give the impression of firmer patterns than shown here.

RESPONSE: We understand the reviewer’s concern and we replaced terms such as ‘latitude’ or ‘latitudinal gradient’ to ‘range location’ or ‘distributional range locations’ throughout the text where applicable. The title indeed excludes the main result for the temperate species, but the present study was done in the light of species range extensions, and hence the main message and novelty lie on the adaptive response of tropical species entering novel environments. The title can only capture so much information, and the revised abstract now clearly delineates the risk-aversion results of the temperate species. Nevertheless, we have also toned down the title stating that ‘Coral reef fishes can become more risk-averse...’ to not generalize the result to all reef fish species.

I also worry about consistency in the interpretation of the patterns that were found. For example, in the abstract you write “Tropical fishes became more risk-averse (increased shelter use and flight initiation distance) and their activity level decreased poleward ... native temperate fishes did not show a consistently altered pattern in their behaviours across latitude.” There was not a consistent increase in risk-aversion within the tropical species (*A. viagiensis* had lower FID at the most poleward site, *A. nigrofuscus* showed no difference in activity levels while the other two species did, and *A. triostegus* showed no increased time inside shelter. Considering that there was evidence of increased risk aversion for *A. strigatus* (activity level declined) and *M. strigatus* showed similar FID changes to *A. viagiensis*, I think that it could be argued that the tropical species also did not show a consistently altered pattern in behaviours across latitude.

RESPONSE: We rewrote the sentence by stating that some tropical fishes did not alter some of their risk-aversion behaviours (only *A. nigrofuscus* for activity level, and *A. triostegus* for time sheltering), but as a general pattern, we can see that tropical fishes showed a consistent alteration in their risk-averse behaviours across range-locations. However, for temperate fishes, only two behaviours were altered (one behaviour for each species). Therefore, there is not a consistent and clear behavioural alteration. We have revised the Abstract to more accurately report the species-specific responses (lines 20-27): “Although some behavioural proxies of risk-aversion remained unaltered for individual tropical fish species, in general tropical fishes became more risk-averse (increased shelter use and/or flight initiation distance), and their activity level decreased at their warm-temperate range locations. Nevertheless, they did not experience a decline in body condition or feeding rate in their novel temperate ranges. Native temperate fishes did not show a consistently altered pattern in their behaviours across their range locations, even though one species increased its flight initiation distance at the warm-temperate location and another one had lowest activity levels at the coldest range location.”

Regarding the FID, it appears there may be an effect of the warm-temperate location, with FID returning to equivalent levels as the subtropical locations for both the tropical and temperate species that were measured at both cold-temperate and subtropical. For that matter, for *A. viagiensis*, the FID at the tropical sites were similar to the warm-temperate location, which may suggest that the FID measured at the subtropical site were anomalously low. While you suggest that decreased FID for *A. viagiensis* at the cold-temperate site may be due to the adoption of other anti-predator strategies (more time in shelter), *M. strigatus* which shows similar reduced FID to *A. viagiensis* does not adopt this behaviour. Regardless of this, I believe your argument for this is not correct, because while spending more time in shelter suggests that fish have adopted extra anti-predator measures, it does not then follow that FID

would be reduced. If anything, I would suggest that increased wariness measured by shelter use should be reflected in higher FID when the fish is away from shelter. If the fishes were moving less far from shelter, this may explain lower FID, but would require an analysis of FID vs distance from shelter to confirm. Finally, considering that there was no difference found in escape distances and escape speeds (line 329-330), suggesting as the authors do on lines 395 -397 that there was an increased flight response at higher latitudes, is not correct across all tropical species, and as mentioned, a similar response was found for at least one temperate species.

RESPONSE: Unfortunately, we cannot confirm this as we did not measure the distance from shelter. However, we modified the argument, including this caveat of our study. See Discussion section (Lines 250-256): “Surprisingly, the tropical species, *A. vaigiensis*, did not show highest flight initiation distance at their most extreme leading range edge (i.e. cold-temperate range location). However, their increased risk-averse behaviour at this range location was shown by the largest time expenditure inside shelters, suggesting that the low flight initiation distance of *A. vaigiensis* at cold-temperate range location could be reflected by their higher proximity to shelters. Yet, the relationship between flight initiation distance and distance to shelter is needed to confirm this argument.” As mentioned in a comment above, the behavioural comparisons are now contrasted among range-locations rather than across a latitudinal gradient. In addition, we clarified the differences in risk-averse behaviours across range locations among the tropical fish species. See abstract, lines 20-27: “Although some behavioural proxies of risk-aversion remained unaltered for individual tropical fish species, in general tropical fishes became more risk-averse (increased shelter use and/or flight initiation distance), and their activity level decreased at their warm-temperate range locations. Nevertheless, they did not experience a decline in body condition or feeding rate in their novel temperate ranges. Native temperate fishes did not show a consistently altered pattern in their behaviours across their range locations, even though one species increased its flight initiation distance at the warm-temperate location and another one had lowest activity levels at the coldest range location.” and discussion, lines 257-259: “Risk-averse behaviours at warm-temperate (all species) and cold-temperate (only for *A. vaigiensis*) range locations did not seem to compromise the fitness of tropical range-extending fishes.”

The evidence is stronger for latitude having an impact on activity levels, but again there is evidence of this for both tropical and temperate species, with one species of each group showing no effect of latitude. The first line of the Discussion, that “We here show that tropical fishes expressed more risk-averse behaviours at their warm-temperate range location than at their subtropical distributional range,” is accurate, but I think that the statement in the conclusion that ‘tropical fishes can show increased risk aversion at the centre of their temperate distributional ranges’ at line 439 – 440 is reaching for a latitudinal explanation which the evidence doesn’t support. If more sites around this warm-temperate location were studied with similar results, that would provide this evidence. At the moment I fear it is a location effect not a latitudinal effect.

RESPONSE: We included a paragraph in the Discussion section about behavioural alterations of temperate fishes across range locations, pointing out that two of their behaviours (FID and activity level) were altered among range-locations. Lines 285-293: “Similar to tropical fishes, one of the studied temperate species (*M. strigatus*) was more risk-averse at the warm-temperate range position, while the other temperate species (*A. strigatus*) was less active at the cold-temperate range position. Hence, our prediction that native temperate fishes would maintain their behavioural responses across distributional ranges, because these species have always naturally occurred there, was not supported by our

findings. At the same time, our study reveals that native temperate fishes did not show a consistently altered pattern in their behaviours across their range locations, as one species increased its flight initiation distance at the warm-temperate location and the second species decreased its activity level at the coldest range location.” **We also specified in the text that not all studied tropical fish species showed increased time sheltering and decreasing activity level across range locations. See Discussion, lines 267-270:** “*Accordingly, two out of three tropical fishes showed increased sheltering behaviours at temperate locations which was associated with highly reduced feeding (i.e. when inside shelter). Hence, the reduced activity levels at temperate locations may be a behavioural strategy adopted by some tropical fishes....*”, **and changed the statement in the Conclusions as suggested by the reviewer, lines 319-321:** “*We reveal that tropical fishes can show more risk-averse behaviours at their warm and cold-temperate distributional ranges under ongoing climate change...*”

The comparisons between tropical and temperate species were useful, and suggest that there are species differences, with much of this difference being driven by *A. vaigiensis* which as mentioned in an earlier review, is quite different from either of the other tropical species. However, latitude does not seem to be having an impact on shelter position, body condition, feeding or bite rate. I stress that I think that their underlying hypothesis may be sound, but that the evidence presented at best only weakly support it.

RESPONSE: Animal behaviour is extremely complex, and behavioural responses to changing environmental conditions are not necessarily expressed through all behavioural proxies. Studies focusing on just one or two proxies might derive erroneous conclusions. Hence, we studied a large range of behavioural proxies. As such, there were proxies that responded in similar ways, and expectedly there were also behaviours that were not affected. We feel that using such a wide range of behavioural proxies provides a much more balanced insights into the complexity of adaptive behavioural responses to climate change, with consistent responses across all proxies not supported by behavioural ecological theory. Nevertheless, we have adjusted our statements as also explained in the above paragraphs.

A more minor critique is that in the introduction there is little discussion of neophobia, which has been heavily studied in fishes and would be relevant to any discussion here. There are multiple studies on risk-induced neophobia in fishes (Joyce et al. 2016; Chivers et al. 2014) In particular, the work on neophobia has suggested that larval fish exposed to high-risk stimuli (such as olfactory cues) may show heightened responses to those or similar cues in later trials. Overall, I do believe that there is merit to the concept that life-stage is an important determinant of predation risk, but this discussion of the influence of life-stage on susceptibility to predation should address neophobia, particularly because of the overall argument that tropicalising species will be introduced to novel environments and predators and the ability to respond would be important for establishment. Indeed, the authors cite several studies on neophobia, which begs the question why this is not addressed directly in the introduction (to be clear, I do not think this should make significant substantial changes in predictions, but it is a missing part of the literature)

RESPONSE: We included and emphasized in the Introduction the concept of neophobia in young fishes and how tropical range-extending fishes can exhibit this behaviour at unknown temperate environments and its possible consequences. See lines 59-71: “*For example, tropical fish species might be particularly vulnerable to a native temperate predator not only because of the lack of eco-evolutionary predator-prey experiences (Sih et al. 2010; Saul & Jeschke 2015) but also because younger fishes are at a greater relative risk of predation (Sogard 1997; Day et al. 2002). Such lack of previous experiences and the*

unknown surrounding environment at temperate locations may induce neophobic predator avoidance in recruits and juvenile tropical fishes (Joyce et al. 2016). Although neophobia can constrain their establishment by reducing foraging opportunities, it would also reduce the potential detrimental costs of the ‘initial encounter’ with a novel predator in a novel environment (Brown et al. 2016). Therefore, learning to assess the risk conditions of unfamiliar environments (Chives et al. 2014) and respond appropriately to novel temperate predators, will allow tropical range extending species to exhibit more efficient predator escape performance behaviour and facilitate their establishment in altered or suboptimal abiotic conditions (Lyon et al. 2007; Djurichkovic et al. 2019).”

Other minor comments

Line 56-57: Think about rewording this slightly – it gave the impression on first reading that the entire species was relieved from historically important predators, not the part of the species/individuals that were expanding their range.

RESPONSE: We changed the sentence to: “Prey species that expand their range to novel environments under climate change are relieved from historically important predators, but at the cost of gaining novel ones (Carthey & Blumstein 2017).” (Lines 57-59).

Line 76: replace ‘can’ with ‘could’ – temperature reducing specifically ‘anti-predator performance’ has not been demonstrated in the work cited. Or provide citation.

RESPONSE: We replaced this as suggested.

Line 80 – 82: I suggest removing this line because it is an oversimplified statement of the influence of life stage for predation susceptibility in fishes.

RESPONSE: Sentence removed as suggested.

Line 94-96: I don’t think it has been demonstrated that early stages of coral reef fishes are more vulnerable to predation due to unfamiliarity to temperate climates and habitats – this has been argued in the introduction, but not demonstrated.

RESPONSE: We rewrote the sentence and cited papers to support our statements. Lines 100-103: “Whilst changes in temperature (Figueira et al, 2019) and habitat context (McCormick et al, 2016) might make early stages of coral reef fishes more vulnerable to predation in temperate environments, shoaling behaviour and refuge proximity in contrast may reduce their risk to temperate predators.”

Figure 2 – can you be state what the asterisked labels mean in the figure.

RESPONSE: We included a sentence in the legend of Figure 2 explaining that “...asterisk indicates when factors such as species or the interaction between range position × species were statistically significant”

Chivers DP, McCormick MI, Mitchell MD, Ramasamy RA, Ferrari MC. Background level of risk determines how prey categorize predators and non-predators. *Proceedings of the Royal Society B: Biological Sciences*. 2014 Jul 22;281(1787):20140355.

RESPONSE: We cited this study as suggested (Lines 67-68).

Joyce BJ, Demers EE, Chivers DP, Ferrari MC, Brown GE. Risk-induced neophobia is constrained by ontogeny in juvenile convict cichlids. *Animal Behaviour*. 2016 Apr 1;114:37-43.

RESPONSE: We cited this study as suggested (Lines 63-65).

Samia DS, Blumstein DT, Stankowich T, Cooper Jr WE. Fifty years of chasing lizards: new insights advance optimal escape theory. *Biological Reviews*. 2016 May;91(2):349-66.

RESPONSE: We cited this study as suggested (Lines 86-88).

Referee 2:

Comments to the Author(s)

The manuscript by Coni and colleagues presents a challenging and integrating in situ experiment to try and understand the real world outcomes for range shifting species when entering a novel system. My first impression of the manuscript is that it takes the work generally done by this team to the next level to attempt to understand the ecological processes for marine vagrant fish. It is complex experiment that the authors have done a good job to try and distil to a clear story. There are some interesting patterns in the results presented that indicate increase risk-aversion. I would love to see more data from the same locations with a greater range of water temperatures as I think it would elucidate what is due to location vs temperature. Just a few ideas and comments from me

I am passionate about a good study design in methods sections, and due to the complexity of the sampling that occurred in the manuscript I think it forms an essential component. In Fig 1 is there not supposed to be line going to the trade-off round circle, I assume from “in situ behaviours”? Should there also be a number(s) in front of shoal size for the continuous explanatory variable?

RESPONSE: We appreciate the reviewer's comments for collecting future data across a larger range of temperatures. We have adjusted Figure 1 following the reviewer's recommendations.

I appreciate the addition of the tropical locations for *A. vaigiensis*. It fascinated me that in the water temperatures for Heron and Magnetic Island were quite close to the sub-tropical and warm-temperate locations (Table S1). So if a response was largely driven by physiological processes linked to temperature, e.g. metabolic rate, then it could explain/support why a difference is not seen between the groups. For example feeding. Do the authors think that this similarity in temperature might be driving the patterns with feeding they saw? I think experimentally the Booth lab would have an idea how temperature changes feeding rate. In cases where a difference is seen, like activity level the responses found here can't be temperature driven, instead that seems to be likely due to the difference in habitat/ecosystem, right?

RESPONSE: This is a good point raised by the reviewer and we added to the Discussion (lines 272-276) that “The water temperatures during the field surveys at the tropical locations in autumn and winter were comparable to those from the subtropical and warm-temperate locations sampled during summer. This supports the notion that any behavioural differences for *A. vaigiensis* among locations were primarily driven by response to a novel environment rather than to seawater temperature per se”

The pattern of body condition getting higher (though surprisingly NS) for *A. vaigiensis* is interesting. Was this an artefact perhaps of SL reducing with cooler locations? I had trouble working out which fish were collected and body condition measured in and what the replication was. I can see that the same fish were used in both observational testing

procedures, some did not give good data for both of the observations I assume. So then I am just not sure how the morphology matches up.

On this if all fish that were observed were collected, why not have size as a continuous covariate, rather than blocked?

RESPONSE: The higher values of Fulton’s Index for *A. vaigiensis* at the warm-temperate and cold-temperate locations were caused by an outlier, as can also be seen from the large error bars. This is the reason why the statistical results were not significant (ns). Additionally, we cannot treat fish size as a continuous variable because we collected these data in size categories. See the supplementary information, ‘Study species’ section. “Since the sizes of the different life stages vary among species, the size categories (mean total length) used were: *A. vaigiensis* (early-juveniles 2.0–4.5 cm, juveniles 4.6–7.0 cm), *A. nigrofuscus* (early-juveniles 5.0–6.0 cm, juveniles 6.1–8.0 cm), *A. triostegus* (early-juveniles 3.0–4.5 cm, juveniles 4.6–6.0 cm), *Atypichthys strigatus* (early-juveniles <4.5 cm, juveniles 4.5–11.0 cm), and *Microcanthus strigatus* (early-juveniles <5.0 cm, juveniles 5.0–9.0 cm), based on Clement (1991), Glasby & Kingsford (1994), Froese & Pauly (2016), Figueira et al. (2009), and Fr  d  rich et al. (2012).”

The individuals collected and used to measure Fulton Condition Index are not the same ones for which the behaviours were quantified. So, we cannot directly associate body condition with the other behaviours because they were not measured on the same individuals. We included this information in the methodology section and also the number of fishes collected per species. See the supplementary information, ‘Body Condition’ section.: “The number of individual fishes collected to calculate the Fulton Condition Index were: *Abudefduf vaigiensis* (tropical = 10, subtropical = 34, warm-temperate = 30, cold-temperate = 45), convict surgeonfish, *Acanthurus triostegus* (subtropical = 5, warm-temperate = 6), *Atypichthys strigatus* (warm-temperate = 8, cold-temperate = 56) and *Microcanthus strigatus* (subtropical = 37, warm-temperate = 13, cold-temperate = 7). Because the surgeonfish *A. nigrofuscus* was not collected, it could not be included in this analysis”.

The link the authors make between risk-aversion and fitness is fine, but it seems that a caveat line should be there at least to admit that these might be the fish that are left, all the other foolish ones may have been eaten. And in the case of the FK condition it may be that there is an optimum that is being selected for.

RESPONSE: Suggestion included. Lines 261-263: “However, this finding might be related to the fact that individuals that did not appropriately responded to novel temperate predators (i.e. allowed closer approach) were less abundant in this study because they had suffered from higher consumption by predators.”

Line 173: This is the first time the convict surgeon species name is mentioned, and since all tropical species genus starts with an “A”, I would give genus in full for specificity.

RESPONSE: Suggestion incorporated.

Line 261: I don’t think sec and min need “.” after

RESPONSE: we deleted the period and opted to write the words out in full.

Line 239-333: Would have been nice to have some values of what the escape speeds and distances were within the main text without having to go look at the Supl. Just a basic range in the brackets where the stats are would be good.

RESPONSE: Information included. Lines 188-193: “All the tropical and temperate fishes maintained their escape distances ($F = 0.825$, $p = 0.576$) and escape speeds ($F = 1.134$, $p =$

0.424) across range locations. The escape distance varied from 2 to 26 cm (mean = ~13 cm) for tropical fishes and 3 to 23 cm (mean = ~12 cm) for temperate species, and the escape speed ranged from 0.22 to 4.88 $\text{cm}\cdot\text{s}^{-1}$ (mean = 0.9 $\text{cm}\cdot\text{s}^{-1}$) and 0.17 to 2.83 $\text{cm}\cdot\text{s}^{-1}$ (mean = 0.8 $\text{cm}\cdot\text{s}^{-1}$), respectively (Fig. S3, Table S2).”

Fig 2, I would have preferred the legend at the top. And maybe something to make the text that indicates the model significant more distinct would be good.

RESPONSE: We moved the legend to the top and we made some changes to highlight species* and the interaction between range location x species* when significant.

At first I was surprised that the Supl Figures not have the same colure scheme. But then I realised this was an amendment in the revision. I would change Supl figures so that they matches main text figures and the map?

RESPONSE: Changed included.

Line 417: Logical rather than logic

RESPONSE: Changed as suggested